# Development of New Pyrazolo [3,4-*b*]Pyridine Derivatives as Potent Anti-Leukemic Agents and Topoisomerase IIα Inhibitors with Broad-Spectrum Cytotoxicity

**DOI:** 10.3390/ph18111770

**Published:** 2025-11-20

**Authors:** Wagdy M. Eldehna, Haytham O. Tawfik, Denisa Veselá, Veronika Vojáčková, Ahmed T. Negmeldin, Zainab M. Elsayed, Taghreed A. Majrashi, Petra Krňávková, Mostafa M. Elbadawi, Moataz A. Shaldam, Ghada H. Al-Ansary, Vladimír Kryštof, Hatem A. Abdel-Aziz

**Affiliations:** 1Department of Pharmaceutical Chemistry, Faculty of Pharmacy, Kafrelsheikh University, Kafrelsheikh 33516, Egypt; mostafa_elbadawi@pharm.kfs.edu.eg (M.M.E.); dr_moutaz_986@pharm.kfs.edu.eg (M.A.S.); 2Department of Pharmaceutical Chemistry, Faculty of Pharmacy, Tanta University, Tanta 31527, Egypt; haytham.omar.mahmoud@pharm.tanta.edu.eg; 3Department of Experimental Biology, Faculty of Science, Palacký University Olomouc, Šlechtitelů 27, 77900 Olomouc, Czech Republic; denisa.vesela@upol.cz (D.V.); veronika.vojackova@upol.cz (V.V.); petra.krnavkova@upol.cz (P.K.); vladimir.krystof@upol.cz (V.K.); 4Department of Pharmaceutical Sciences, College of Pharmacy and Thumbay Research Institute for Precision Medicine, Gulf Medical University, Ajman 4184, United Arab Emirates; 5Department of Pharmaceutical Organic Chemistry, Faculty of Pharmacy, Cairo University, Cairo 12613, Egypt; 6Scientific Research and Innovation Support Unit, Faculty of Pharmacy, Kafrelsheikh University, Kafrelsheikh 6860404, Egypt; 7Department of Pharmacognosy, College of Pharmacy, King Khalid University, Asir 62521, Saudi Arabia; tamajrashi@kku.edu.sa; 8Department of Pharmaceutical Chemistry, Faculty of Pharmacy, Ain Shams University, Abassia, Cairo 11566, Egypt; ghada.mohamed@pharma.asu.edu.eg; 9Institute of Molecular and Translational Medicine, Faculty of Medicine and Dentistry, Palacký University Olomouc, Hněvotínská 5, 77900 Olomouc, Czech Republic; 10Applied Organic Chemistry Department, National Research Center, Dokki, Cairo 12622, Egypt

**Keywords:** TOPIIα, molecular modeling, Western blotting, apoptosis, pyrazolopyridine

## Abstract

**Background/Objectives:** In the current medical era, Topoisomerase II is recognized as an essential enzyme that regulates DNA topology during critical biological processes such as DNA replication, transcription, and repair. This study aimed to design, synthesize, and biologically evaluate a new series of pyrazolo[3,4-*b*]pyridines (**8a**–**g**, **10a**–**g**, and **12**) as potential anticancer agents and Topoisomerase II inhibitors. **Methods:** The synthesized compounds were subjected to in vitro anticancer screening at the National Cancer Institute (NCI, USA). Active derivatives were further evaluated through a five-dose screening to determine their antiproliferative potency. Selected compounds were examined for their effects on leukemia cell lines (K562 and MV4-11), and mechanistic studies were performed to assess DNA damage, cell cycle distribution, and apoptosis-related protein modulation. Additionally, enzyme inhibition assays were conducted to determine Topoisomerase IIα (TOPIIα) inhibition. **Results:** Initial single-dose screening identified several active compounds, notably **8b**, **8c**, **8e**, **8f**, **10b**, **10c**, **10e**, and **10f**. Among these, compound **8c** exhibited potent and broad-spectrum antiproliferative activity across the NCI cancer cell line panel, with a GI_50_ MG-MID value of 1.33 µM (range: 0.54–2.08 µM). The synthesized molecules showed moderate to good anti-leukemic efficacy against K562 and MV4-11 cells. Mechanistic investigations revealed that compound **8c** induced DNA damage and S-phase cell cycle arrest, leading to apoptosis as evidenced by the modulation of PARP-1, Bax, XIAP, and Caspases. Furthermore, target-based assays confirmed that compound **8c** significantly inhibited the DNA relaxation activity of TOPIIα in a dose-dependent manner, comparable to etoposide. **Conclusions:** The study highlights compound **8c** as a promising pyrazolo[3,4-*b*]pyridine derivative with potent antiproliferative activity and effective inhibition of Topoisomerase IIα. These findings suggest its potential as a lead scaffold for further optimization in anticancer drug development..

## 1. Introduction

Cancer is still one of the leading causes of death worldwide and remains a serious public health concern [1,2]. By 2050, the global burden of cancer is anticipated to reach 19 million cancer diagnoses worldwide, with an estimated 10.5 million cancer-related deaths [3]. Among the diverse types of cancer, leukemia stands out as one of the most aggressive and potentially fatal hematological malignancies [4]. Acute myelogenous leukemia (AML), chronic myelogenous leukemia (CML), acute lymphocytic leukemia (ALL), and chronic lymphocytic leukemia (CLL) are the four primary subtypes of leukemia that are clinically distinguished based on cell lineage and the rate of disease progression [5]. Each subtype exhibits distinct pathological, genetic, and clinical features, which require individualized treatment approaches. Although leukemia often responds well to chemotherapy, the range of safe and durable treatment options remains limited [6]. As a result, the development of novel, more effective therapies for leukemia remains a pressing and continuous task. Pyrazolo [3,4-*b*]pyridine is a privileged fused heterocyclic scaffold commonly found in pharmaceuticals and drug-like substances [7]. Research has shown that derivatives of pyrazolo [3,4-*b*]pyridine (Figure 1) exhibit anticancer properties through various mechanisms, including inhibition of several targets such as cyclin-dependent kinases (**I**–**IV**) [8,9,10], tubulin polymerization (**V**) [11], glycogen synthase kinase-3 (**VI**) [12], hematopoietic cell kinase (**VII**) [13], fibroblast growth factor receptor (**VIII**) [14], tropomyosin receptor kinases (**IX**) [15] and TOPII (**X**–**XI**) [16,17].

Indole is a simple chemical scaffold with a bicyclic planar structure found in many natural products [18]. The biological actions of indole derivatives include antibacterial [19], antiinflammatory [20], anticancer [21,22], anticonvulsant [23], antiviral [24], antidiabetic [25], and antioxidant [26] properties. Regarding cancer proliferation inhibition, indole derivatives are highly effective because they strongly target multiple pathways (Figure 2), including PIM (**XII**) [27], CDK (**XIII**) [28], EGFR (**XIV**) [29], PI3Kα (**XV**) [30], Bcl-2/Mcl-1 (**XVI**) [31], TOPII (**XV**–**XVI**) [32], etc. [33].

The endonuclease class enzymes known as DNA topoisomerases (TOPs) are widely distributed throughout all life spheres and play essential roles in fundamental cellular processes [34,35]. TOP enzymes carry out a variety of tasks linked to the maintenance of DNA metabolism and regulate DNA topology during transcription and replication [36,37]. Based on their structure, amino acid sequence, catalytic activity, and reaction mechanism, TOPs can be divided into type I and type II [38,39]. Numerous investigations have demonstrated that TOPs are overexpressed in tumor cells, and their inhibition suppressed tumor cell growth, establishing TOPs as valuable targets for the development of anticancer therapeutics [40,41]. Furthermore, leukemia’s higher topoisomerase IIα expression levels have been associated with increased sensitivity to agents targeting this enzyme [42]. TOPII inhibitors, such as doxorubicin and etoposide, have been extensively employed in cancer therapy. Although these compounds demonstrate clinical efficacy, their application is often limited by toxicity and adverse side effects [43,44]. Doxorubicin, for instance, is known to induce dose-dependent cardiotoxicity primarily through the generation of reactive oxygen species (ROS) and mitochondrial dysfunction, leading to cardiomyocyte apoptosis [45,46]. Etoposide, on the other hand, is associated with myelosuppression due to its interference with DNA synthesis and repair in rapidly dividing hematopoietic cells [47]. Moreover, resistance to TOPII inhibitors in leukemia patients remains a significant clinical challenge. Mechanisms contributing to drug resistance include the deletion or downregulation of the TOPIIα gene, upregulation of DNA repair enzymes such as PARP and RAD51, and enhanced drug efflux via ATP-binding cassette (ABC) transporters [48]. These adaptations reduce drug efficacy and complicate treatment outcomes, thus, the development of novel and more efficient TOPII inhibitors is the subject of extensive research [41,49].

TOPII inhibitors are known for their inflexible, coplanar aromatic structures, which serve as templates for additional structural optimization to facilitate the intercalation of stable ternary complexes with DNA. The molecular scope of inhibitor design tactics has been expanded beyond “poison inhibition” to encompass “catalytic inhibition”, “dual-targeting”, and “structure-induced blockade” [50]. As a result, new TOPII inhibitors with non-classical structures are appearing in preclinical research. A tendency towards structural and mechanistic diversity is reflected in these structures, which extend beyond conventional scaffolds and incorporate a variety of chemotypes [50]. Furthermore, the creation of dual-target compounds that integrate TOPII inhibitors with other target inhibitors, such as PARP [51], HDAC [52], and tubulin [53], results in long-lasting lethal stress and reduces resistance.

Previous studies have highlighted planar structures as a key pharmacophoric feature of TOPII inhibitors such as the scaffolds found in compounds **X** [16] and **XI** [17], both of which contain pyrazolo [3,4-*b*]pyridine in their structure. Additionally, the indole ring, due to its planar structure, can act as an intercalation site for DNA interaction, as demonstrated by compounds **XV** and **XVI**, which showed promising TOPII inhibition activities. All things considered, the compounds in this study were based on a hybridization strategy involving pyrazolo [3,4-*b*]pyridine as the main scaffold bearing the indole moiety at three varying positions to achieve promising activity against TOPII. As a part of the structure, phenyl rings were also grafted to afford binding interaction with the lipophilic DNA groove binding regions (Figure 3). The obtained compounds were tested in a preliminary anticancer screening at the National Cancer Institute (NCI), and the most promising were subjected to biological mechanistic studies and molecular docking to explore their potential anticancer mechanisms.

## 2. Results and Discussion

### 2.1. Chemistry

Figure 1 and Figure 2 show the synthetic routes taken to produce the target pyrazolo [3,4-*b*]pyridine derivatives (**8a**–**g**, **10a**–**g**, and **12**). To obtain 3-(1*H*-indol-3-yl)-3-oxopropanenitrile (**2**), indole (**1**) was heated with cyanoacetic acid, and acetic anhydride was involved [54].

Additionally, 3-oxo-3-phenylpropanenitrile (**4**) and 3-(1*H*-indol-3-yl)-3-oxopropanenitrile (**2**) were condensed with phenylhydrazine in refluxing absolute ethanol to create the 3-substituted-1-phenyl-1*H*-pyrazol-5-amines (**5** and **9**) [55,56,57,58,59,60]. As shown in Figure 1 and Figure 2, the target pyrazolo [3,4-*b*]pyridine derivatives (**8a**–**g**, **10a**–**g**, and **12**) were then prepared using a one-pot, three-component reaction] that included the relevant 3-substituted-1-phenyl-1*H*-pyrazol-5-amine (**5** or **9**), an equimolar amount of the corresponding 3-oxo-3-arylpropanenitrile (**2** or **4**), and the appropriate aldehyde (**3a**-**g** and **11**).

Intermediates **6a**–**g** are produced as part of the reaction mechanism, and they are cyclized and then water-eliminated to produce intermediates **7a**–**g**. The required compounds **8a**–**g** are subsequently obtained through a final oxidative dehydrogenation process, in which atmospheric oxygen acts as the oxidant. Nuclear magnetic resonance (NMR) spectroscopy (^1^H and ^13^C) (Appendix A) and high-resolution mass spectrometry (HRMS) (Appendix A) validated the structures of the synthesized compounds, which were in agreement with the proposed structures.

### 2.2. Structure Elucidation of the Target Compounds

The ^1^H NMR spectra of target compounds **8a**–**g**, **10a**–**g**, and **12** confirmed their structures. In particular, the absence of the aldehydic proton (CH=O) signals from aldehydes (**3a**–**g** and **11**) and the active methylene protons (CH_2_) from nitriles (**2** and **4**) around δ 3.5 ppm, as well as the disappearance of characteristic signals around δ 5.0 ppm, which correspond to the two NH_2_ protons of the precursor 3-substituted-1-phenyl-1*H*-pyrazol-5-amines (**5** and **9**). The addition of aromatic moieties throughout the process was further supported by the spectra, which showed an increase in aromatic proton signals.

The suggested structures were further supported by the ^13^C NMR spectra, which revealed the elimination of carbon signals associated with nitrile and aldehyde carbonyl groups, typically detected at δ 190 ppm. These modifications verified that the initial materials had completely changed into the final pyrazolo [3,4-*b*]pyridine derivatives.

With variances falling within the permissible range of ±0.4%, the elemental analysis findings showed good agreement with the estimated values for the target compounds’ molecular formulae. The structures were further validated by molecular ion peaks obtained from high-resolution mass spectrometry (HRMS) that matched the predicted molecular weights. The effectiveness and selectivity of the synthetic processes were further demonstrated by high-performance liquid chromatography (HPLC) analysis, which verified that all synthesized compounds had a purity higher than 95.00% (Appendix A).

### 2.3. Biological Evaluation

The NCI 60 human cancer cell line panel was used for preliminary in vitro anticancer screening at a single dose of 10 μM.

The United States’ National Cancer Institute (NCI) conducted the first anticancer assessment of the recently synthesized indole-conjugated pyrazolo [3,4-*b*]pyridine derivatives (**8a**–**g**, **10a**–**g**, and **12**). The comprehensive NCI 60 human tumor cell line panel, which comprises leukemia, non-small cell lung, melanoma, central nervous system (CNS), ovarian, renal, prostate, and breast cancer cell lines, was first screened in vitro at a fixed concentration of 10 µM. The findings, presented as the mean percentage growth (G%) of treated cells compared to untreated controls, shed light on the cytotoxic effects (G% less than 0) as well as the cytostatic activity (G% between 0 and 100).

The single-dose activity profiles of compounds **8a**–**g, 10a**–**g,** and **12** were examined using the COMPARE algorithm on 60 distinct cancer cell lines. The pyrazolo [3,4-*b*]pyridines assessed exhibited a broad range of cytotoxic effects on various cancer types, with anticancer activities ranging from weak to very potent [61]. Table 1 summarizes the growth inhibition percentages (GI%) for each target molecule, which are computed as (100–G%).

According to the NCI criteria, derivatives **8d, 8g,** and **10d** (mean GI% values below 20) did not exhibit significant cytotoxicity against the examined cell lines (except for **8d** towards NCI-H226 and **10d** towards CCRF-CEM, HL-60(TB), and MOLT-4) [62]. Compounds **8a** (mean GI% = 24), **10a** (mean GI% = 43), **10g** (mean GI% = 29), and **12** (mean GI% = 23) exposed promising sensitivity towards limited representative cell lines (leukemia; K-562 and renal cancer; RXF 393) (Table 1).

The remaining eight compounds; 3-hydroxyphenyl derivatives (**8b** and its counterpart **10b**), 4-hydroxyphenyl derivatives (**8c** and its counterpart **10c**), 4-hydroxy-3-methoxyphenyl derivatives (**8e** and its counterpart **10e**) and 3-hydroxy-4-methoxyphenyl derivatives (**8f** and its counterpart **10f**) showed auspicious results in NCI preliminary screening with mean GI% = 120, 154, 138, 125, 126, 118, 138 and 122 (Table 1).

The antiproliferative activity of compounds **8a** (mean GI% = 24) and **10a** (mean GI% = 43) was considerably increased by adding a hydroxyl group to the *meta* position of one of the phenyl rings. This resulted in **8b** and **10b** derivatives, with mean growth inhibition (GI%) values of 120% and 126%, respectively. Among the 60 cell lines examined, both compounds showed the most potent anticancer effects. According to Table 1, each demonstrated fatal activity against 42 cell lines, 33 of which were shared by both, and inhibitory effects on the other lines. The GI% values for these lines ranged from 46 to 99.

When the hydroxyl group in compound **8b** was shifted from the *meta* to the *para* position, obtaining compound **8c**, the antiproliferative activity increased. Both derivatives exerted a lethal effect on 47 cell lines. The same structure modification of **10b**, yielding compound **10c**, led to a modest decrease in activity, and **10c** exerted a lethal effect only on 37 cell lines. The mean GI% values for **8c** and **10c** were 154% and 118%, respectively (Table 1).

Introducing a methoxy group at the *para* position adjacent to the hydroxyl group in compounds **8b** and **10b** resulted in enhanced antiproliferative activity, as observed in the obtained derivatives 8e and 10e, both of which showed a mean GI% of 138. These compounds exerted lethal effects on 39 common cancer cell lines. Lastly, compounds **8e** and **10e** showed enhanced anticancer activity with mean GI% values of 125 and 122, respectively, when the hydroxyl and methoxy groups were repositioned, with the hydroxyl in the *para* position and the methoxy in the *meta* position, as in compounds **8f** and **10f** with 34 and 35 cancer cell lines were fatally affected by these compounds, respectively.

Based on their in vitro cytotoxic characteristics against the NCI-60 human cancer cell line panel, the structure–activity relationship (SAR) of the synthesized compounds is depicted in Figure 4. The mean growth inhibition percentages (GI%) presented in Table 1 serve as the basis for this analysis. Important pharmacophoric characteristics that contribute to potency and selectivity across several cancer cell types were identified by establishing a correlation between the observed anticancer activity and the structural alterations within the pyrazolo [3,4-*b*]pyridine scaffold. This SAR analysis lays the groundwork for future structural optimization in the development of more potent anticancer drugs, providing insightful information on how specific substituents affect biological activity.

In the next step, the most potent derivatives, including 3-hydroxyphenyl derivatives (**8b** and its counterpart **10b**), 4-hydroxyphenyl derivatives (**8c** and its counterpart **10c**), 4-hydroxy-3-methoxyphenyl derivatives (**8e** and its counterpart **10e**) and 3-hydroxy-4-methoxyphenyl derivatives (**8f** and its counterpart **10f**), were screened against the 60 cancer cell lines at 10-fold dilutions and five different concentrations (0.01, 0.1, 1, 10 and 100 μM) [61]. Following established experimental procedures, the sulforhodamine B (SRB) colorimetric assay was used to evaluate cell viability [63]. Cell density and vitality after chemical treatment are indirectly assessed by measuring cellular protein content in this experiment. The molar concentrations of the substances needed to inhibit 50% of cell proliferation (GI_50_) and induce 50% cell death (LC_50_) are represented by the GI_50_ and LC_50_ values for each cancer cell line that is tested. Table 2 summarizes these important pharmacological parameters, providing information on the cytotoxic and antiproliferative efficacy of the investigated agents [64,65]. The NCI 60 cancer cell line panel demonstrated strong antiproliferative activity across all chosen compounds (**8b**, **8c**, **8e**, **8f**, **10b**, **10c**, **10e**, and **10f**), with efficacy ranging from sub-micromolar to low micromolar concentrations, according to the results. These results demonstrate compounds’ strong growth inhibition against a wide range of cancer cell types, underscoring their considerable potential as potent anticancer agents.

The results showed that most compounds had significant inhibitory effects, with GI_50_ values ranging from 0.15 µM to 5 µM. Only a few cell lines showed reduced sensitivity, with GI_50_ values not exceeding 15 µM. Among the tested compounds, **8c** (a 4-hydroxyphenyl derivative) and **8e** (a 4-hydroxy-3-methoxyphenyl derivative) demonstrated remarkable potency, particularly against leukemia, followed by CNS, renal, and breast cancer cell lines.

Compound **8c** displayed outstanding activity with GI_50_ values ranging from 0.15 to 4.43 μM among the tested NCI panel, including 15 cell lines with GI_50_ below 1.00 μM. Interestingly, it showed sub-micromolar inhibitory activity towards all leukemia, all CNS (except SF-268 and SNB-19), renal (786-0, RXF 393, and UO-31), and breast cancer cell lines (MCF7 and HS 578T). Regarding compound **8e**, it displayed remarkable activity with GI_50_ values ranging from 0.23 to 4.68 μM (with 10 cancer cell lines with GI_50_ below 1.00 μM). Compound **8e** showed sub-micromolar inhibitory activity towards all leukemia (except K-562 and MOLT-4), all CNS (except SF-268 and SNB-19), and renal cancer cell lines (786-0 and RXF 393) (Table 2).

Additionally, an average activity metric for all cell lines is provided by the mean graph midpoint (MG-MID). The compounds **8b**, **8c**, **8e**, **8f**, **10b**, **10c**, **10e,** and **10f** showed MID (average sensitivity of all cell lines in µM) values of 2.43, 1.33, 1.93, 3.04, 3.37, 4.23, 2.65, and 2.49 µM, respectively (Table 3).

The ratio of the entire panel mean graph midpoint (MID) to the individual subpanel MID was used to calculate the selectivity of these substances. The subpanel MID shows the average sensitivity of cell lines within a particular tumor subpanel, whereas the whole panel MID shows the average sensitivity of all cell lines to the test agent. Compounds with selectivity ratios below this cutoff are regarded as nonselective, whilst those with ratios above 2 may suggest a preference for specific tumor types. With selectivity ratios ranging from 0.54 to 2.46, the drugs in this investigation showed broad-spectrum anticancer efficacy across the nine tumor subpanels examined. With an average MID of 1.33 μM, compound **8c** demonstrated the highest potency overall and the most effective and selective treatment of leukemia (MID = 0.54 μM, selectivity = 2.46). Likewise, compound 8e had significant potency and selectivity against leukemia, attaining a MID of 0.97 μM and a selectivity ratio of 1.98, with an average MID of 1.93 μM.

### 2.4. Antiproliferative Activity Against Leukemia Cells

Following the NCI 60 screening, the synthesized derivatives (**8b**, **8c**, **8e**, **8f**, **10b**, **10c**, **10e,** and **10f**) were further evaluated for their antiproliferative activity against leukemia cancer cell lines, including chronic myeloid leukemia (K562) and acute monocytic leukemia (MV4-11), as shown in Table 4.

With the exception of compound **8e**, the chosen pyrazolo [3,4-*b*]pyridine derivatives efficiently suppressed the proliferation of MV4-11 cells, with GI_50_ values ranging from 0.72 to 9.03 µM. With a GI_50_ value of 0.72 µM, pyrazolo [3,4-*b*]pyridine **8c** exhibited the strongest sub-micromolar cytotoxic effect. Furthermore, with GI_50_ values of 3.55 and 3.70 µM, respectively, compounds **8b** and **8f** demonstrated low single-digit micromolar cytotoxicity against the MV4-11 cell line. The remaining compounds (**10b**, **10c**, **10e**, and **10f**), on the other hand, demonstrated substantial single-digit micromolar cytotoxic activity, as indicated by Table 4, with GI_50_ values ranging from 8.03 to 9.03 µM.

The pyrazolo [3,4-*b*]pyridine derivatives showed GI_50_ values ranging from 0.72 to 6.52 µM for the K562 leukemia cell line. Compound **8c** once more exhibited exceptional sub-micromolar cytotoxicity (GI_50_ = 0.72 µM). With GI_50_ values ranging from 2.50 to 6.52 µM, compounds **8b**, **8f**, **10b**, **10c**, **10e**, and **10f** demonstrated single-digit micromolar cytotoxic action. However, as Table 4 summarizes, compound 8e exhibited noticeably less activity, with a GI_50_ value exceeding 10 µM.

Overall, SAR analysis indicates that subtle structural modifications have a marked influence on biological activity, notably enhancing potency against leukemia cancer cell lines. Among the tested compounds, **8c** exhibited the highest potency and was therefore selected for further experiments to elucidate the molecular mechanism of action of the presented derivatives.

### 2.5. Immunodetection and Cell Cycle Analysis of MV4-11

Based on the promising antiproliferative activity observed in the initial screening and SAR analysis, further experiments were conducted to characterize the molecular mechanisms underlying the cytotoxic effects of the most potent compound **8c**. Asynchronously growing MV4-11 cells were incubated with increasing concentrations of **8c** (1.25, 2.5, 5.0, and 10 µM) for 24 h and subsequently analyzed by immunodetection (Figure 5) and cell cycle flow cytometry (Figure 6).

Treatment with **8c** efficiently induced the cleavage of initiator caspase-9, which was accompanied by cleavage of executioner caspases −3 and −7. Once activated, they cleave a variety of cellular substrates, including poly(ADP-ribose) polymerase 1 (PARP-1), thereby promoting commitment to cell death [66]. The presence of the cleaved 89 kDa fragment of PARP-1 clearly indicates a concentration-dependent progression of cell death in MV4-11 cells following **8c** treatment. This was further associated with a slight increase in the proapoptotic protein Bax and, conversely, a decrease in the antiapoptotic protein XIAP. In addition, **8c** induced phosphorylation of histone H2AX at serine 139 (γ-H2AX), indicative of DNA damage, which is likely the trigger responsible for the initial activation of caspases [67,68] (Figure 5).

To further support these findings, flow cytometry analysis of cell cycle distribution was performed in MV4-11 cells treated with **8c** for 24 h, using propidium iodide staining. Treatment with **8c** resulted in a dose-dependent decrease in the S-phase cell population, accompanied by a reduction in the number of cells progressing to the G_2_/M phase and a significant increase in the sub-G_1_ population. These changes are indicative of replication stress during the S-phase and corroborate the strong induction of cell death previously observed by immunoblotting (Figure 6).

### 2.6. Topoisomerase Relaxation Assay

Finally, we sought to investigate the underlying mechanism responsible for the replication stress induced by compound **8c**, leading to cell death. Replication stress is often associated with inhibiting TOPs, enzymes essential for maintaining DNA topology during replication. Notably, several established TOP inhibitors, such as camptothecin (a TOPI inhibitor) and etoposide (a TOPII inhibitor), possess planar aromatic structures that facilitate DNA intercalation [44]. Given the planar aromatic character of the synthesized compound series, we hypothesized that **8c** might interfere with the activity of TOPI or TOPII.

To test this hypothesis, we performed topoisomerase relaxation assays to evaluate the inhibitory potential of **8c** on the ability of TOPI and TOPIIα to relax supercoiled plasmid DNA. Reaction products were separated by agarose gel electrophoresis and visualized using GelRed nucleic acid stain. Camptothecin (CPT) and etoposide were employed as positive controls for TOPI and TOPIIα inhibition, respectively. **8c** did not inhibit TOPI-mediated DNA relaxation; however, it impaired the relaxation activity of TOPIIα in a concentration-dependent manner (Figure 7).

### 2.7. Kinase Assays

The newly prepared pyrazolo [3,4-*b*]pyridines (**8b**, **8c**, **8e**, **8f**, **10b**, **10c**, **10e**, and **10f**) underwent further evaluation against a focused kinase panel (Abl, FLT3-ITD, and PDGFR) to assess off-target effects and confirm TOPIIα selectivity. At concentrations up to 10 µM, these molecules exhibited minimal inhibitory activity (Appendix A) across the tested kinases, demonstrating no clinically relevant off-target interactions. Furthermore, we intend to extend the profiling to include a broader kinase panel in our future work, which will allow for a thorough assessment of kinome-wide selectivity.

### 2.8. Docking

The structure of **8c** was subjected to molecular docking modelling in order to assess the potential mode of binding inside TOPII. As anticipated from the best docking pose, **8c** fits well into the major groove region of the DNA double helix (Figure 8). The intercalation was observed for the cocrystal with the DNA bases DC8 and DC13, while **8c** was observed with DC8, DC9, DC12, and DC13. The π-interactions were observed between these bases and the aromatic systems of **8c,** including the main scaffold, one of the phenyl rings, the indole ring, and the phenolic ring. Additionally, the cyano group formed a hydrogen bond with Gln778, and a similar interaction occurred with one of the oxygen atoms of the cocrystal ligand. In addition, lipophilic interactions with Met782 and Arg503 were also observed with **8c**. Furthermore, the OH group in the para position enabled sulfur-x interaction with Met782 and improved the π-stacking interaction with DT9. The docking score of **8c** (−10.4 kcal/mol) was lower than that of the cocrystal ligand (−14.7 kcal/mol) while maintaining the important interaction patterns.

## 3. Materials and Methods

### 3.1. Chemistry

#### 3.1.1. General

Commercial suppliers provided all of the materials, which were used exactly as supplied. Silica gel plates (Merck KGaA, Darmstadt, Germany) were subjected to TLC in order to track the purity and development of the reaction. Uncorrected melting points were measured using a Stuart SMP30 apparatus (Cole-Parmer Ltd., Stone, Staffordshire, UK), and standard equipment was used for elemental analysis using a Vario EL III elemental analyzer (Elementar Analysensysteme GmbH, Langenselbold, Germany), FTIR using a PerkinElmer Spectrum Two FTIR spectrometer (Waltham, MA, USA), and NMR (^1^H, ^13^C, and DEPT-135) were obtained on a Bruker Avance III HD 400 MHz spectrometer (Bruker BioSpin GmbH, Rheinstetten, Germany). A Bruker MicroTOF spectrometer was used to record high-resolution mass spectra (Bruker Daltonics, Billerica, MA, USA). As stated in the literature, starting ingredients **2** [54], **5** [55], and **9** [55] were synthesized. A ZORBAX Eclipse Plus C18 column (4.6 × 150 mm, 5 µm; Agilent Technologies, Santa Clara, CA, USA) was used for HPLC analysis, which verified that the produced chemicals were more than 95% pure.

#### 3.1.2. General Procedure for Synthesis of Pyrazolo [3,4-b]Pyridines (**8a**–**g**, **10a**–**g**, and **12**)

TLC was used to track the reaction progress of a combination of different aldehydes (**3a**–**g** and **11**) (0.0015 mole), equimolar quantities of 3-oxo-3-arylpropanenitriles (**2** and **4**) (0.0015 mole), and 3-substituted-1-phenyl-1*H*-pyrazol-5-amines (**5** and **9**) (0.0015 mole) that were refluxed in 20 mL of absolute ethanol for 12 h. The reaction mixture was finished and then left to cool to room temperature. Filtration, air drying, and recrystallization of the resultant precipitate from ethanol produced the required very pure pyrazolo [3,4-*b*]pyridine derivatives (**8a**–**g**, **10a**–**g**, and **12**).

#### 3.1.3. 6-(1H-Indol-3-yl)-1,3,4-Triphenyl-1H-Pyrazolo [3,4-b]Pyridine-5-Carbonitrile (**8a**)

Yield (85%) as a white powder, with Mp: 245 °C. HPLC: R_T_ 7.098 min (purity: 96.00%). ^1^H NMR (500 MHz, DMSO*d*_6_) δ (ppm): 11.68 (d, *J* = 2.9 Hz, 1H, NH), 7.81–7.78 (m, 3H, Arom-H), 7.63–7.57 (m, 4H, Arom-H), 7.49–7.39 (m, 3H, Arom-H), 7.33–7.29 (m, 6H, Arom-H), 7.20–7.16 (m, 2H, Arom-H), 7.13 (tt, *J* = 7.2, 2.3 Hz, 2H, Arom-H). ^13^C NMR (101 MHz, DMSO*d*_6_) δ (ppm): 149.7, 145.6, 145.3, 139.1, 138.2, 136.3, 134.5, 130.7, 129.8, 129.3, 129.1, 128.9, 128.3, 127.4, 127.2, 125.7, 125.2, 123.7, 122.4, 122.2, 121.4, 120.0, 111.8, 108.1, 99.4, 85.7. HRMS (ESI): *m*/*z*: [M + H]^+^ calcd. 488.1870 and found 488.1862. Anal. Calcd. (Found) For C_33_H_21_N_5_: C, 81.29 (81.15); H, 4.34 (4.35); N, 14.36 (14.29)%.

#### 3.1.4. 4-(3-Hydroxyphenyl)-6-(1H-Indol-3-yl)-1,3-Diphenyl-1H-Pyrazolo [3,4-b]Pyridine-5-Carbonitrile (**8b**)

Yield (83%) as a white powder, with Mp: 232 °C. HPLC: R_T_ 5.770 min (purity: 99.40%). ^1^H NMR (500 MHz, DMSO*d*_6_) δ (ppm): 11.68 (d, *J* = 2.9 Hz, 1H, NH), 9.80 (s, 1H, OH), 7.81–7.78 (m, 3H, Arom-H), 7.72–7.54 (m, 7H, Arom-H), 7.21–7.07 (m, 5H, Arom-H), 6.83–6.74 (m, 2H, Arom-H), 6.71 (t, *J* = 2.1 Hz, 1H, Arom-H), 6.59 (dd, *J* = 8.1, 2.4 Hz, 1H, Arom-H). ^13^C NMR (101 MHz, DMSO*d*_6_) δ (ppm): 158.5, 150.5, 144.8, 143.8, 139.3, 136.0, 134.1, 130.8, 129.7, 128.6, 128.1, 128.0, 127.3, 124.0, 121.5, 119.0, 115.2, 114.6, 112.9, 102.1. HRMS (ESI): *m*/*z*: [M + H]^+^ calcd. 504.1819 and found 504.1810. Anal. Calcd. (Found) For C_33_H_21_N_5_O: C, 78.71 (78.87); H, 4.20 (4.18); N, 13.91 (13.86)%.

#### 3.1.5. 4-(4-Hydroxyphenyl)-6-(1H-Indol-3-yl)-1,3-Diphenyl-1H-Pyrazolo [3,4-b]Pyridine-5-Carbonitrile (**8c**)

Yield (89%) as a white powder, with Mp: 295 °C. HPLC: R_T_ 5.165 min (purity: 98.53%). ^1^H NMR (500 MHz, DMSO*d*_6_) δ (ppm): 11.92 (s, 1H, NH), 9.85 (s, 1H, OH), 8.53–8.44 (m, 2H, Arom-H), 8.39–8.29 (m, 2H, Arom-H), 7.69–7.61 (m, 2H, Arom-H), 7.57 (dd, *J* = 8.3, 3.3 Hz, 1H, Arom-H), 7.49–7.43 (m, 1H, Arom-H), 7.31–7.25 (m, 2H, Arom-H), 7.23–7.11 (m, 7H, Arom-H), 6.71–6.56 (m, 2H, Arom-H). ^13^C NMR (126 MHz, DMSO*d*_6_) δ (ppm): 159.2, 156.3, 153.7, 151.0, 147.5, 138.9, 136.9, 132.1, 131.5, 130.1, 129.7, 129.4, 128.5, 128.0, 127.4, 126.4, 124.8, 123.1, 122.3, 122.0, 121.4, 119.7, 115.2, 113.3, 112.7, 110.8, 100.3. ^13^C NMR-DEPT-135 (126 MHz, DMSO*d*_6_) δ: 131.5 (↑), 130.1 (↑), 129.7 (↑), 129.4 (↑), 128.5 (↑), 128.0 (↑), 127.4 (↑), 123.1 (↑), 122.3 (↑), 122.0 (↑), 121.4 (↑), 115.2 (↑), 112.7 (↑). HRMS (ESI): *m*/*z*: [M + H]^+^ calcd. 504.1819 and found 504.1809. Anal. Calcd. (Found) For C_33_H_21_N_5_O: C, 78.71 (78.59); H, 4.20 (4.23); N, 13.91 (13.97)%.

#### 3.1.6. 4-(2-Hydroxy-3-Methoxyphenyl)-6-(1H-Indol-3-yl)-1,3-Diphenyl-1H-Pyrazolo [3,4-b]Pyridine-5-Carbonitrile (**8d**)

Yield (85%) as a white powder, with Mp: 271 °C. HPLC: R_T_ 7.888 min (purity: 99.41%). ^1^H NMR (400 MHz, DMSO*d*_6_) δ (ppm): 11.33 (s, 1H, NH), 9.41 (s, 1H, OH), 8.45–6.76 (m, 18H, Arom-H), 3.88 (s, 3H, OCH_3_). ^13^C NMR (101 MHz, DMSO*d*_6_) δ (ppm): 160.3, 153.2, 150.0, 149.7, 147.1, 143.0, 139.1, 138.3, 136.2, 130.5, 129.9, 129.8, 128.9, 127.8, 127.5, 126.9, 126.2, 122.4, 121.7, 120.5, 118.3, 116.9, 112.5, 112.0, 106.4, 102.5, 56.2. Anal. Calcd. (Found) For C_34_H_23_N_5_O_2_: C, 76.53 (76.38); H, 4.34 (4.30); N, 13.13 (13.09)%.

#### 3.1.7. 4-(3-Hydroxy-4-Methoxyphenyl)-6-(1H-Indol-3-yl)-1,3-Diphenyl-1H-Pyrazolo [3,4-b]Pyridine-5-Carbonitrile (**8e**)

Yield (91%) as a white powder, with Mp: 266 °C. HPLC: R_T_ 7.413 min (purity: 96.44%). ^1^H NMR (500 MHz, DMSO*d*_6_) δ (ppm): 11.92 (s, 1H, NH), 9.13 (s, 1H, OH), 8.48 (d, *J* = 7.9 Hz, 2H, Arom-H), 8.36–8.33 (m, 2H, Arom-H), 7.67–7.63 (m, 2H, Arom-H), 7.57 (d, *J* = 8.0 Hz, 1H, Arom-H), 7.48–7.45 (m, 1H, Arom-H), 7.28 (ddt, *J* = 8.1, 5.6, 1.9 Hz, 2H, Arom-H), 7.22 (td, *J* = 7.5, 1.2 Hz, 1H, Arom-H), 7.18–7.11 (m, 4H, Arom-H), 6.82 (d, *J* = 2.1 Hz, 1H, Arom-H), 6.78 (d, *J* = 8.3 Hz, 1H, Arom-H), 6.73 (dd, *J* = 8.2, 2.1 Hz, 1H, Arom-H), 3.77 (s, 3H, OCH_3_). ^13^C NMR (126 MHz, DMSO*d*_6_) δ (ppm): 156.2, 153.4, 150.9, 149.3, 147.4, 146.8, 138.9, 136.9, 132.1, 130.1, 129.7, 129.2, 128.6, 127.9, 127.4, 126.9, 126.4, 123.1, 122.4, 122.0, 121.4, 119.5, 117.0, 113.3, 112.7, 112.2, 110.8, 100.3, 56.3. ^13^C NMR-DEPT-135 (126 MHz, DMSO*d*_6_) δ: 130.1 (↑), 129.7 (↑), 129.2 (↑), 128.6 (↑), 127.9 (↑), 127.4 (↑), 123.1 (↑), 122.4 (↑), 122.0 (↑), 121.4 (↑), 117.0 (↑), 112.7 (↑), 112.2 (↑), 56.3 (↑). HRMS (ESI): *m*/*z*: [M + H]^+^ calcd. 534.1925 and found 534.1922. Anal. Calcd. (Found) For C_34_H_23_N_5_O_2_: C, 76.53 (76.65); H, 4.34 (4.32); N, 13.13 (13.16)%.

#### 3.1.8. 4-(4-Hydroxy-3-Methoxyphenyl)-6-(1H-Indol-3-yl)-1,3-Diphenyl-1H-Pyrazolo [3,4-b]Pyridine-5-Carbonitrile (**8f**)

Yield (87%) as a white powder, with Mp: 239 °C. HPLC: R_T_ 7.355 min (purity: 98.55%). ^1^H NMR (500 MHz, DMSO*d*_6_) δ (ppm): 11.92 (d, *J* = 2.9 Hz, 1H, NH), 9.40 (s, 1H, OH), 8.49 (d, *J* = 2.7 Hz, 1H, Arom-H), 8.47 (d, *J* = 7.9 Hz, 1H, Arom-H), 8.36–8.31 (m, 2H, Arom-H), 7.68–7.63 (m, 2H, Arom-H), 7.57 (d, *J* = 8.0 Hz, 1H, Arom-H), 7.49–7.45 (m, 1H, Arom-H), 7.31–7.25 (m, 2H, Arom-H), 7.22 (td, *J* = 7.4, 1.2 Hz, 1H, Arom-H), 7.17 (d, *J* = 4.4 Hz, 4H, Arom-H), 7.05 (dd, *J* = 8.1, 2.1 Hz, 1H, Arom-H), 6.82 (d, *J* = 8.1 Hz, 1H, Arom-H), 6.74 (d, *J* = 2.0 Hz, 1H, Arom-H), 3.34 (s, 3H, OCH_3_). ^13^C NMR (126 MHz, DMSO*d*_6_) δ (ppm): 156.4, 153.5, 151.0, 148.6, 147.5, 147.4, 138.9, 136.9, 132.4, 130.2, 129.7, 129.4, 128.6, 127.9, 127.4, 126.4, 125.0, 123.1, 123.0, 122.4, 122.0, 121.4, 119.8, 115.5, 114.9, 113.3, 112.7, 110.7, 100.2, 55.7. ^13^C NMR-DEPT-135 (126 MHz, DMSO*d*_6_) δ: 130.2 (↑), 129.7 (↑), 129.4 (↑), 128.6 (↑), 127.9 (↑), 127.4 (↑), 123.1 (↑), 123.0 (↑), 122.4 (↑), 122.0 (↑), 121.4 (↑), 115.5 (↑), 114.9 (↑), 112.7 (↑), 55.7 (↑). HRMS (ESI): *m*/*z*: [M + H]^+^ calcd. 534.1925 and found 534.1912. Anal. Calcd. (Found) For C_34_H_23_N_5_O_2_: C, 76.53 (76.67); H, 4.34 (4.38); N, 13.13 (13.18)%.

#### 3.1.9. 4-(3,4-Dimethoxyphenyl)-6-(1H-Indol-3-yl)-1,3-Diphenyl-1H-Pyrazolo [3,4-b]Pyridine-5-Carbonitrile (**8g**)

Yield (88%) as a white powder, with Mp: 228 °C. HPLC: R_T_ 10.632 min (purity: 99.55%). ^1^H NMR (500 MHz, DMSO*d*_6_) δ (ppm): 12.26 (s, 1H, NH), 8.44 (s, 2H, Arom-H), 8.22–8.18 (m, 4H, Arom-H), 7.81 (d, *J* = 2.2 Hz, 2H, Arom-H), 7.73 (dd, *J* = 8.5, 2.1 Hz, 2H, Arom-H), 7.56 (d, *J* = 7.1 Hz, 2H), 7.28 (td, *J* = 7.6, 1.5 Hz, 4H, Arom-H), 7.19 (d, *J* = 8.5 Hz, 2H, Arom-H), 3.88 (s, 3H, OCH_3_), 3.84 (s, 3H, OCH_3_). ^13^C NMR (126 MHz, DMSO*d*_6_) δ (ppm): 181.9, 153.1, 152.9, 149.1, 137.1, 135.8, 126.7, 126.3, 125.4, 123.9, 122.8, 121.9, 119.1, 114.2, 113.3, 112.9, 112.3, 108.4, 56.3, 56.0. ^13^C NMR-DEPT-135 (126 MHz, DMSO*d*_6_) δ: 152.9 (↑), 135.8 (↑), 126.3 (↑), 123.9 (↑), 122.8 (↑), 121.9 (↑), 113.3 (↑), 112.9 (↑), 112.3 (↑), 56.3 (↑), 56.0 (↑). HRMS (ESI): *m*/*z*: [M + H]^+^ calcd. 548.2081 and found 548.2080. Anal. Calcd. (Found) For C_35_H_25_N_5_O_2_: C, 76.77 (76.91); H, 4.60 (4.57); N, 12.79 (12.89)%.

#### 3.1.10. 3-(1H-Indol-3-yl)-1,4,6-Triphenyl-1H-Pyrazolo [3,4-b]Pyridine-5-Carbonitrile (**10a**)

Yield (86%) as a white powder, with Mp: 280 °C. HPLC: R_T_ 7.001 min (purity: 98.27%). ^1^H NMR (500 MHz, DMSO*d*_6_) δ (ppm): 11.22–11.19 (m, 1H, NH), 8.38 (d, *J* = 8.0 Hz, 2H, Arom-H), 8.20 (d, *J* = 7.7 Hz, 1H, Arom-H), 8.02 (dd, *J* = 6.6, 2.9 Hz, 2H, Arom-H), 7.68–7.61 (m, 6H, Arom-H), 7.59 (s, 1H, Arom-H), 7.56–7.53 (m, 2H, Arom-H), 7.48 (t, *J* = 7.6 Hz, 2H, Arom-H), 7.42 (t, J = 7.4 Hz, 1H, Arom-H), 7.35 (d, J = 7.8 Hz, 1H, Arom-H), 7.17–7.12 (m, 2H, Arom-H). ^13^C NMR (126 MHz, DMSO*d*_6_) δ (ppm): 160.4, 153.1, 150.0, 142.8, 139.0, 138.2, 136.1, 135.0, 130.6, 130.5, 129.9, 129.9, 129.9, 129.0, 128.9, 127.3, 127.1, 126.2, 122.4, 121.8, 121.4, 120.5, 118.1, 112.4, 112.0, 106.3, 102.4. ^13^C NMR-DEPT-135 (126 MHz, DMSO*d*_6_) δ: 130.6 (↑), 130.5 (↑), 129.9 (↑), 129.9 (↑), 129.9 (↑), 129.0 (↑), 128.9 (↑), 127.3 (↑), 127.1 (↑), 122.4 (↑), 121.8 (↑), 121.4 (↑), 120.5 (↑), 112.0 (↑). HRMS (ESI): *m*/*z*: [M + H]^+^ calcd. 488.1870 and found 488.1868. Anal. Calcd. (Found) For C_33_H_21_N_5_: C, 81.29 (81.42); H, 4.34 (4.37); N, 14.36 (14.41)%.

#### 3.1.11. 4-(3-Hydroxyphenyl)-3-(1H-Indol-3-yl)-1,6-Diphenyl-1H-Pyrazolo [3,4-b]Pyridine-5-Carbonitrile (**10b**)

Yield (83%) as a white powder, with Mp: 247 °C. HPLC: R_T_ 5.736 min (purity: 99.79%). ^1^H NMR (500 MHz, DMSO*d*_6_) δ (ppm): 11.32 (d, *J* = 2.9 Hz, 1H, NH), 9.88 (s, 1H, OH), 8.40 (d, *J* = 8.0 Hz, 2H, Arom-H), 8.03–8.01 (m, 2H, Arom-H), 7.68–7.62 (m, 6H, Arom-H), 7.39–7.32 (m, 3H, Arom-H), 7.20–7.13 (m, 3H, Arom-H), 7.04–7.02 (m, 1H, Arom-H), 6.98 (d, *J* = 7.6 Hz, 1H, Arom-H), 6.96 (d, *J* = 2.1 Hz, 1H, Arom-H). HRMS (ESI): *m*/*z*: [M+H]^+^ calcd. 504.1819 and found 504.1824. Anal. Calcd. (Found) For C_33_H_21_N_5_O: C, 78.71 (78.90); H, 4.20 (4.19); N, 13.91 (13.94)%.

#### 3.1.12. 4-(4-Hydroxyphenyl)-3-(1H-Indol-3-yl)-1,6-Diphenyl-1H-Pyrazolo [3,4-b]Pyridine-5-Carbonitrile (**10c**)

Yield (85%) as a white powder, with Mp: 259 °C. HPLC: R_T_ 5.054 min (purity: 99.32%). ^1^H NMR (500 MHz, DMSO*d*_6_) δ (ppm): 11.33 (d, *J* = 2.8 Hz, 1H, NH), 10.07 (s, 1H, OH), 8.38 (d, *J* = 8.1 Hz, 2H, Arom-H), 8.24–8.22 (m, 1H, Arom-H), 8.02–8.00 (m, 2H, Arom-H), 7.65–7.61 (m, 5H, Arom-H), 7.42 (s, 1H, Arom-H), 7.37 (dd, *J* = 8.6, 6.9 Hz, 3H, Arom-H), 7.20–7.09 (m, 3H, Arom-H), 6.86–6.83 (m, 2H, Arom-H). HRMS (ESI): *m*/*z*: [M + H]^+^ calcd. 504.1819 and found 504.1817. Anal. Calcd. (Found) For C_33_H_21_N_5_O: C, 78.71 (78.86); H, 4.20 (4.16); N, 13.91 (13.95)%.

#### 3.1.13. 4-(2-Hydroxy-3-Methoxyphenyl)-3-(1H-Indol-3-yl)-1,6-Diphenyl-1H-Pyrazolo [3,4-b]Pyridine-5-Carbonitrile (**10d**)

Yield (90%) as a white powder, with Mp: 277 °C. HPLC: R_T_ 7.845 min (purity: 99.81%). ^1^H NMR (400 MHz, DMSO*d*_6_) δ (ppm): 11.67 (s, 1H, NH), 9.21 (s, 1H, OH), 8.27 (d, *J* = 7.6 Hz, 1H, Arom-H), 7.88 (d, *J* = 2.5 Hz, 1H, Arom-H), 7.81–7.71 (m, 3H, Arom-H), 7.60 (t, *J* = 7.9 Hz, 3H, Arom-H), 7.51–7.42 (m, 3H, Arom-H), 7.32–7.30 (m, 1H, Arom-H), 7.21–7.14 (m, 5H, Arom-H), 6.95 (s, 1H, Arom-H), 3.82 (s, 3H, OCH_3_). ^13^C NMR (101 MHz, DMSO*d*_6_) δ (ppm): 163.3, 150.1, 148.9, 148.8, 148.4, 139.4, 137.0, 129.6, 127.6, 125.4, 124.6, 124.5, 123.4, 122.1, 121.4, 120.2, 120.1, 119.7, 116.3, 112.2, 109.3, 91.9, 56.3. HRMS (ESI): *m*/*z*: [M + H]^+^ calcd. 534.1925 and found 534.1918. Anal. Calcd. (Found) For C_34_H_23_N_5_O_2_: C, 76.53 (76.66); H, 4.34 (4.32); N, 13.13 (13.16)%.

#### 3.1.14. 4-(3-Hydroxy-4-Methoxyphenyl)-3-(1H-Indol-3-yl)-1,6-Diphenyl-1H-Pyrazolo [3,4-b]Pyridine-5-Carbonitrile (**10e**)

Yield (93%) as a white powder, with Mp: 249 °C. HPLC: R_T_ 7.380 min (purity: 99.95%). ^1^H NMR (500 MHz, DMSO*d*_6_) δ (ppm): 11.35 (d, *J* = 2.9 Hz, 1H, NH), 9.45 (s, 1H, OH), 8.39 (d, *J* = 8.0 Hz, 2H, Arom-H), 8.33–8.30 (m, 1H, Arom-H), 8.03–7.99 (m, 2H, Arom-H), 7.67–7.62 (m, 5H, Arom-H), 7.55–7.35 (m, 3H, Arom-H), 7.17 (tt, *J* = 7.1, 5.4 Hz, 2H, Arom-H), 7.05 (d, *J* = 8.3 Hz, 1H, Arom-H), 6.98 (d, *J* = 2.2 Hz, 1H, Arom-H), 6.95 (dd, *J* = 8.2, 2.2 Hz, 1H, Arom-H), 3.89 (s, 3H, OCH_3_). ^13^C NMR (126 MHz, DMSO*d*_6_) δ (ppm): 160.4, 153.3, 150.0, 149.7, 147.1, 143.0, 139.1, 138.3, 136.2, 130.6, 129.9, 129.9, 129.6, 129.0, 128.6, 127.8, 127.5, 127.0, 126.2, 122.4, 121.8, 121.7, 121.2, 120.6, 118.3, 116.9, 112.5, 112.4, 112.0, 106.4, 102.6, 56.2. ^13^C NMR-DEPT-135 (126 MHz, DMSO*d*_6_) δ: 130.6 (↑), 129.9 (↑), 129.9 (↑), 129.0 (↑), 127.8 (↑), 127.0 (↑), 122.4 (↑), 121.8 (↑), 121.7 (↑), 121.2 (↑), 120.6 (↑), 116.9 (↑), 112.5 (↑), 112.0 (↑), 56.2 (OCH_3_ ↑). HRMS (ESI): *m*/*z*: [M + H]^+^ calcd. 534.1925 and found 534.1929. Anal. Calcd. (Found) For C_34_H_23_N_5_O_2_: C, 76.53 (76.38); H, 4.34 (4.28); N, 13.13 (13.08)%.

#### 3.1.15. 4-(4-Hydroxy-3-Methoxyphenyl)-3-(1H-Indol-3-yl)-1,6-Diphenyl-1H-Pyrazolo [3,4-b]Pyridine-5-Carbonitrile (**10f**)

Yield (88%) as a white powder, with Mp: 276 °C. HPLC: R_T_ 7.283 min (purity: 99.42%). ^1^H NMR (500 MHz, DMSO*d*_6_) δ (ppm): 11.34 (d, *J* = 2.8 Hz, 1H, NH), 9.61 (s, 1H, OH), 8.38 (d, *J* = 8.3 Hz, 2H, Arom-H), 8.17 (d, *J* = 7.8 Hz, 1H, Arom-H), 8.04–7.99 (m, 2H, Arom-H), 7.67–7.59 (m, 5H, Arom-H), 7.45–7.36 (m, 2H, Arom-H), 7.15 (dt, *J* = 21.7, 7.2 Hz, 2H, Arom-H), 7.05 (dt, *J* = 8.1, 1.8 Hz, 1H, Arom-H), 7.01 (d, *J* = 1.7 Hz, 1H, Arom-H), 6.88 (dd, *J* = 8.0, 1.4 Hz, 1H, Arom-H), 6.16 (t, *J* = 2.0 Hz, 1H, Arom-H), 3.38 (s, 3H, OCH_3_). ^13^C NMR (126 MHz, DMSO*d*_6_) δ (ppm): 160.6, 153.4, 150.1, 148.9, 147.7, 142.9, 139.0, 138.3, 136.1, 130.6, 130.0, 129.8, 128.9, 127.7, 127.0, 126.2, 125.1, 123.2, 122.3, 121.8, 121.2, 120.5, 118.6, 115.7, 114.9, 112.4, 111.9, 106.5, 102.5, 55.9. ^13^C NMR-DEPT-135 (126 MHz, DMSO*d*_6_) δ: 130.6 (↑), 130.0 (↑), 129.8 (↑), 128.9 (↑), 127.7 (↑), 127.0 (↑), 123.3 (↑), 122.3 (↑), 121.8 (↑), 121.2 (↑), 120.5 (↑), 115.7 (↑), 114.9 (↑), 111.9 (↑), 55.9 (OCH_3_ ↑). HRMS (ESI): *m*/*z*: [M + H]^+^ calcd. 534.1925 and found 534.1920. Anal. Calcd. (Found) For C_34_H_23_N_5_O_2_: C, 76.53 (76.37); H, 4.34 (4.33); N, 13.13 (13.11)%.

#### 3.1.16. 4-(3,4-Dimethoxyphenyl)-3-(1H-Indol-3-yl)-1,6-Diphenyl-1H-Pyrazolo [3,4-b]Pyridine-5-Carbonitrile (**10g**)

Yield (91%) as a white powder, with Mp: 284 °C. HPLC: R_T_ 9.795 min (purity: 99.32%). ^1^H NMR (500 MHz, DMSO*d*_6_) δ (ppm): 11.30 (d, *J* = 2.8 Hz, 1H, NH), 8.38 (d, *J* = 8.0 Hz, 2H, Arom-H), 8.12 (d, *J* = 7.9 Hz, 1H, Arom-H), 8.05–8.00 (m, 2H, Arom-H), 7.69–7.61 (m, 5H, Arom-H), 7.41 (dd, *J* = 20.8, 7.7 Hz, 2H, Arom-H), 7.19–7.01 (m, 5H, Arom-H), 6.12 (d, *J* = 2.7 Hz, 1H, Arom-H), 3.85 (s, 3H, OCH_3_), 3.36 (s, 3H, OCH_3_). ^13^C NMR (126 MHz, DMSO*d*_6_) δ (ppm): 160.6, 153.1, 150.7, 150.1, 148.7, 142.9, 139.0, 138.3, 136.1, 130.6, 130.0, 129.9, 129.0, 127.6, 127.1, 126.6, 126.2, 122.9, 122.3, 121.8, 121.1, 120.4, 118.5, 114.3, 112.5, 112.0, 111.9, 106.4, 102.4, 56.2, 55.7. ^13^C NMR-DEPT-135 (126 MHz, DMSO*d*_6_) δ: 130.6 (↑), 130.0 (↑), 129.9 (↑), 129.8 (↑), 129.0 (↑), 127.6 (↑), 127.1 (↑), 122.9 (↑), 122.3 (↑), 121.8 (↑), 121.1 (↑), 120.4 (↑), 114.3 (↑), 112.0 (↑), 111.9 (↑), 56.2 (OCH_3_ ↑), 55.7 (OCH_3_ ↑). HRMS (ESI): *m*/*z*: [M + H]^+^ calcd. 548.2081 and found 548.2074 Anal. Calcd. (Found) For C_35_H_25_N_5_O_2_: C, 76.77 (76.59); H, 4.60 (4.62); N, 12.79 (12.85)%.

#### 3.1.17. 4-(1H-Indol-3-yl)-1,3,6-Triphenyl-1H-Pyrazolo [3,4-b]Pyridine-5-Carbonitrile (**12**)

Yield (89%) as a white powder, with Mp: 257 °C. HPLC: R_T_ 7.166 min (purity: 99.25%). ^1^H NMR (500 MHz, DMSO*d*_6_) δ (ppm): 11.62 (d, *J* = 2.8 Hz, 1H, NH), 8.35–8.29 (m, 2H, Arom-H), 8.09–8.03 (m, 2H, Arom-H), 7.66–7.61 (m, 5H, Arom-H), 7.46–7.39 (m, 4H, Arom-H), 7.18–7.09 (m, 4H, Arom-H), 7.02–6.94 (m, 3H, Arom-H). ^13^C NMR (126 MHz, DMSO*d*_6_) δ (ppm): 161.4, 150.7, 147.7, 147.6, 138.8, 138.4, 136.2, 132.2, 130.6, 130.0, 129.8, 129.0, 128.9, 128.7, 127.7, 127.4, 126.2, 122.4, 122.3, 120.4, 120.0, 118.9, 112.7, 112.2, 109.0, 102.9. ^13^C NMR-DEPT-135 (126 MHz, DMSO*d*_6_) δ: 130.6 (↑), 130.0 (↑), 129.8 (↑), 129.0 (↑), 128.9 (↑), 128.9 (↑), 128.7 (↑), 127.7 (↑), 127.4 (↑), 122.4 (↑), 122.3 (↑), 120.4 (↑), 120.0 (↑), 112.2 (↑). HRMS (ESI): *m*/*z*: [M + H]^+^ calcd. 488.18697 and found 488.18698. Anal. Calcd. (Found) For C_33_H_21_N_5_: C, 81.29 (81.44); H, 4.34 (4.32); N, 14.36 (14.41)%.

### 3.2. Biological Evaluation

#### 3.2.1. In Vitro Antitumor Screening Against 60 Cancer Cell Lines

In accordance with the NCI’s usual technique, the synthetic compounds’ in vitro anticancer activity was assessed against a panel of 60 human tumor cell lines from nine different tissue types (Appendix A). To evaluate each compound’s potency and cytotoxic profile, key dose–response parameters, including GI_50_, TGI, and LC_50_, were computed [69,70,71,72,73,74,75,76,77], and detailed data are provided in the Appendix A.

#### 3.2.2. Cell Lines

Recognized cell banks provided the MV4-11 and K562 cancer cell lines, which were then cultivated at 37 °C in a humidified CO_2_ incubator under standard conditions in RPMI-1640 or DMEM medium supplemented with fetal bovine serum and antibiotics, and detailed data are provided in the Appendix A.

#### 3.2.3. Cell Viability Assay

In order to calculate GI_50_ values from dose–response curves using fluorescence measurements, cells were seeded in 96-well plates and treated with different doses of the test substances for 72 h. This was followed by a viability assessment based on resazurin, and detailed data are provided in the Appendix A.

#### 3.2.4. Flow Cytometry

The test substance was applied at different concentrations to asynchronously developing cells, which were then harvested after 24 h, fixed in ice-cold 70% ethanol, and incubated on ice for 30 min. Cell cycle distribution was examined by flow cytometry with a 488 nm laser after PBS washing and propidium iodide staining, and ModFit LT software (version 5.0.9) was used to quantify the results, and detailed data are provided in the Appendix A.

#### 3.2.5. Immunoblotting

RIPA buffer was used to generate cell lysates, and proteins were separated using SDS-PAGE before being transferred to nitrocellulose membranes. Following blocking, membranes were incubated with certain primary antibodies overnight at 4 °C. Peroxidase-conjugated secondary antibodies were then added, and SuperSignal West Pico reagents and a LAS-4000 CCD camera were used for detection, and detailed data are provided in the Appendix A.

#### 3.2.6. Topoisomerase Relaxation Assay

Supercoiled pBR322 plasmid and the corresponding assay buffers were used for the Topoisomerase I and IIα assays. The assays were then incubated for 30 min at 37 °C at 350 rpm. Complete experimental details are included in the Appendix A. Reactions were halted, products were separated using 5% agarose gel electrophoresis, and they were then seen using GelRed staining and a FLA-7000 digital image analyzer.

#### 3.2.7. Molecular Docking

The RCSB-PDB site was used to obtain the coordinates of TOPII (PDB: 3QX3 [78]). The three-dimensional structures of **8c** and the cocrystal ligand were prepared and optimized using Marvin Sketch [79] as part of the docking investigation. AutoDock Vina [80] was used to carry out docking operations. The active site’s coordinates (x, y, z) were 32.9/95.4/50.8 with a size 24.5/20.2/17.5. The 2D schematic presentation and 3D visualization were created using the Discovery Studio 2021 client [81].

## 4. Conclusions

The synthesized pyrazolo [3,4-*b*]pyridine derivatives demonstrated antiproliferative activity, particularly against leukemia-derived cancer cell lines, with **8c** emerging as the most potent candidate. Detailed mechanistic studies revealed that **8c** induces DNA damage leading to cell death in MV4-11 cells, as evidenced by caspase activation, PARP-1 cleavage, and modulation of pro- and antiapoptotic proteins. Additionally, **8c** caused S-phase cell cycle perturbations indicative of replication stress and confirmed strong induction of cell death. Topoisomerase relaxation assays identified TOPIIα as a potential molecular target, with **8c** inhibiting its activity in a concentration-dependent manner. These findings highlight **8c** as a promising lead compound for the further development of anticancer agents targeting TOPIIα-associated pathways.

## Data Availability

The original contributions presented in the study are included in the article and Appendix A, further inquiries can be directed to the corresponding authors.

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
