# Peer review of "Pharmaceuticals2025, 18(11), 1770;https://doi.org/10.3390/ph18111770"

_pharmaceuticals, 2025, doi:10.3390/ph18111770_

Round 1

Reviewer 1 Report

Comments and Suggestions for Authors

The present manuscript reports the search of novel anticancer agents in the series of pyrazolo[3,4-b]pyridine derivatives. The design of target compounds is adequately rationalized. The newly obtained pyrazolo[3,4-b]pyridines are properly characterized; their structure and purity are not in doubt. According to NCI screening, most of the studied compounds exhibited antiproliferative or toxic activity against a broad panel of cancer cells. For the most active compound the properties of Topoisomerase II inhibitor were demonstrated, that was in agreement with the result of molecular docking studies.

The manuscript can be accepted for publishing after the following minor issues are addressed.

1) Reference should be given for the key three-component reaction, not only for the synthesis of starting pyrazol-5-amines.

2) What is the oxidant at the last step of the discussed mechanism?

3) Line 164. It is not clear, what the phrase “The hydrazone compounds that were evaluated…” refers to. It seems to have no connection with the course of discussion.

4) In Table 4 no data for positive control is given.

5) When describing docking studies, the role of OH-group, which is important for activity according to SAR, is worth mentioning.

6) In General procedure the quantity of the reagents (in mole) should be given.

7) Melting point 245 oC is reported for all compounds.

8) In description of 13C NMR spectra chemical shifts should be given to one decimal place (152.1 not 152.08 ppm).

9) In Supplementary file Figures S6, S7, S22, S23 are absent.

Author Response

The manuscript can be accepted for publishing after the following minor issues are addressed:

  1. Reference should be given for the key three-component reaction, not only for the synthesis of starting pyrazol-5-amines.

Response: We appreciate the reviewer’s comment. The references for the key three-component reaction have been included in the revised manuscript to clarify its literature basis.

  1. What is the oxidant at the last step of the discussed mechanism?

Response: We thank the reviewer for this important and insightful comment.
The final step in the proposed mechanism (conversion of intermediates 7a-g to 8a-g) involves an oxidative dehydrogenation under aerobic conditions. This transformation typically proceeds via molecular oxygen (air) acting as the oxidant. Similar oxidative conversions have been reported to occur spontaneously upon exposure to air oxygen, which serves as a cheap, green, and readily available oxidant for such transformations. Molecular oxygen promotes oxidative dehydrogenation by converting C–N and C–O bonds into C=N and C=O bonds, respectively. This mechanistic feature has been described in related studies (Behbehani et al., ACS Omega 2019, 4, 15794–15802, https://doi.org/10.1021/acsomega.9b02430; ACS Omega 2019, 4, 13293–13300, https://doi.org/10.1021/acsomega.9b01650; and Molecules 2022, 27, 497, https://doi.org/10.3390/molecules27020497; Org. Biomol. Chem. 2024, http://dx.doi.org/10.1039/D4OB02045F). Accordingly, we have clarified in the revised manuscript that atmospheric oxygen serves as the oxidant in this step.

  1. Line 164. It is not clear, what the phrase “The hydrazone compounds that were evaluated…” refers to. It seems to have no connection with the course of discussion.

Response: We thank the reviewer for pointing out this ambiguity. The sentence has been revised to clarify the meaning and ensure that it is directly connected to the context of the discussion.

  1. In Table 4 no data for positive control is given.

Response: We greatly appreciate the reviewer’s suggestion. Data for two reference compounds tested in parallel were added to Table 4.

  1. When describing docking studies, the role of OH-group, which is important for activity according to SAR, is worth mentioning.

Response: Thank you for bringing this to our attention. The interaction involving the OH-group was amended to the manuscript and was highlighted.

  1. In General procedure the quantity of the reagents (in mole) should be given.

Response: We thank the reviewer for the helpful suggestion. The number of moles of all reagents has been included in the General Procedure section as requested.

  1. Melting point 245 oC is reported for all compounds.

Response: We thank the reviewer for noticing this mistake. The melting point value of 245 °C was a typographical error and has been corrected. The correct melting points for all compounds have now been provided in the revised manuscript.

  1. In description of 13C NMR spectra chemical shifts should be given to one decimal place (152.1 not 152.08 ppm).

Response: We highly appreciate the reviewer’s request. The 13C NMR chemical shift values have been revised to one decimal place throughout the manuscript, as suggested.

  1. In Supplementary file Figures S6, S7, S22, S23 are absent.

Response: We thank the reviewer for pointing out this omission. The missing Figures have now been included in the revised Supplementary Information file.

Reviewer 2 Report

Comments and Suggestions for Authors

This paper focuses on the synthesis, anti-leukemia activity, and TOPIIα inhibitory mechanism of pyrazolo[3,4-b]pyridine derivatives. Through experiments such as NCI 60 cell screening, cell cycle analysis, Western blot, and molecular docking, it is confirmed that compound 8c has excellent in vitro antiproliferative activity and TOPIIα inhibitory ability. It provides a potential lead compound for the research and development of anti-leukemia drugs and has certain academic value and application prospects.

  1. The introduction only outlines the research basis of TOPII inhibitors and pyrazolo[3,4-b]pyridine derivatives, without integrating key advancements in the field from 2023 to 2025 (such as structural optimization strategies for TOPIIα-selective inhibitors and mechanism studies on pyrazolopyridine compounds reversing leukemia drug resistance). It fails to reflect the connection between this research and current cutting-edge directions, thereby weakening the positioning of the research's innovativeness.
  2. Pyrazolopyrimidine and pyrazolo[3,4-b]pyridine have very similar structures. Please add a section on the inhibitor design concept based on the principle of bioisosterism in the design to enhance the reliability of rational design, and provide several examples of Pyrazolopyrimidine, such as: DOI: 10.2174/1570180820666230602093051; https://doi.org/10.1007/s00044-022-02861-7.
  3. Although it is mentioned that clinically used TOPII inhibitors such as doxorubicin and etoposide have toxicity, it does not specify the specific types of toxicity (e.g., cardiotoxicity of doxorubicin, myelosuppression of etoposide) and their molecular mechanisms. It also fails to elaborate on the drug resistance rates and resistance mechanisms (e.g., TOPIIα gene deletion, upregulation of DNA repair enzymes) in leukemia patients to existing drugs in clinical practice. This results in an insufficient sense of urgency for the motivation of this study, which is "developing new anti-leukemia agents". Please add the reference PMID: 33372884 to the paragraph "Limitations of Clinical Application of TOPII Inhibitors" in the introduction section.
  4. Compound 8c induces ROS/apoptosis. Please provide some literature explanations, such as the unique properties of heterocyclic compounds (PMID: 33026807, 33085980).
  5. Introduction: "Research Progress of TOPII Inhibitors" The research prospects in the paragraph or conclusion section explain the trend of combined application of TOPII inhibitors or the possibility of combining with inhibitors of other targets. (https://doi.org/10.1002/adtp.201900202)
  6. When detecting apoptosis-related proteins such as PARP-1, Bax, and XIAP, only qualitative band diagrams were provided. No grayscale normalization (such as ImageJ quantitative analysis) was performed on the β-actin internal reference protein, and no statistical data from 3 independent repeated experiments were provided.
  7. Figure 4 shows that "the activity of the para-hydroxy group (8c) is higher than that of the meta-hydroxy group (8b)", but it does not explain this phenomenon in combination with molecular docking results: for example, whether the para-hydroxy group forms a stronger hydrogen bond with Gln778 at the active site of TOPIIα, or whether it improves the π-stacking interaction between the molecule and DNA bases. It only describes that "structural modification affects activity" without revealing the molecular essence of the structure-activity relationship.
  8. Although the π-interactions between 8c and DNA bases as well as the hydrogen bond with Gln778 are mentioned, the key binding sites of co-crystal ligands (such as etoposide) are not compared (e.g., the electrostatic interaction between etoposide and Asp479, and the hydrogen bond with DNA phosphate groups). Thus, it is impossible to clarify whether 8c retains the core binding characteristics of TOPII inhibitors or whether there exists a unique mode of action (such as binding to both TOPIIα and DNA simultaneously).
  9. In Figure 1, compound V does not list obvious target information. In compounds X and XI, "TOPO II" should be "TOP II".
  10. In Line 151, the author mentions that the HPLC purity of all compounds is above 95%, but only the purity information of compound 8c is available in the SI.
  11. The paragraph marker for Line 154 is missing.
  12. Table 1 shows a good foundation of work, and it would be more convincing if there were SD values. The symbol appearing in the annotation (line 213) does not appear in the table; instead, there is an unannotated symbol in the table. Table 2 has the same issue.
  13. The article lacks a discussion section.
  14. The structural confirmation information and HRMS of compound 8d are missing in SI. (Figure S6 and S7)
    10d (Figure S22 and S23)
    13C NMR of 8a and 8b
    HRMS of 12

Author Response

  1. The introduction only outlines the research basis of TOPII inhibitors and pyrazolo[3,4-b]pyridine derivatives, without integrating key advancements in the field from 2023 to 2025 (such as structural optimization strategies for TOPIIα-selective inhibitors and mechanism studies on pyrazolopyridine compounds reversing leukemia drug resistance). It fails to reflect the connection between this research and current cutting-edge directions, thereby weakening the positioning of the research's innovativeness.

Response: We highly appreciate the reviewer’s positive critique. The Introduction has been revised to include recent advances (2023–2025) on TOPIIα-selective inhibitors and pyrazolopyridine-based mechanisms addressing leukemia drug resistance.

  1. Pyrazolopyrimidine and pyrazolo[3,4-b]pyridine have very similar structures. Please add a section on the inhibitor design concept based on the principle of bioisosterism in the design to enhance the reliability of rational design, and provide several examples of Pyrazolopyrimidine, such as: DOI: 2174/1570180820666230602093051; https://doi.org/10.1007/s00044-022-02861-7.

Response: We appreciate the reviewer’s comment. Indeed, we do not have any pyrazolopyrimidine-based derivatives in the current manuscript.

  1. Although it is mentioned that clinically used TOPII inhibitors such as doxorubicin and etoposide have toxicity, it does not specify the specific types of toxicity (e.g., cardiotoxicity of doxorubicin, myelosuppression of etoposide) and their molecular mechanisms. It also fails to elaborate on the drug resistance rates and resistance mechanisms (e.g., TOPIIα gene deletion, upregulation of DNA repair enzymes) in leukemia patients to existing drugs in clinical practice. This results in an insufficient sense of urgency for the motivation of this study, which is "developing new anti-leukemia agents". Please add the reference PMID: 33372884 to the paragraph "Limitations of Clinical Application of TOPII Inhibitors" in the introduction section.

Response: We thank the reviewer for this helpful comment. The Introduction has been revised to specify the toxicity types of doxorubicin and etoposide, elaborate on resistance mechanisms in leukemia, and include the suggested reference.

  1. Compound 8c induces ROS/apoptosis. Please provide some literature explanations, such as the unique properties of heterocyclic compounds (PMID: 33026807, 33085980).

Response: We thank the reviewer for this valuable comment and for suggesting relevant literature. In response, we have cited a related study that supports the observed ROS-mediated apoptotic activity of our compound. The newly added reference discusses the mechanistic role of heterocyclic scaffolds in inducing oxidative stress and apoptosis, which is consistent with our findings and the structural features of compound 8c. The citation has been included in the revised manuscript.

  1. Introduction: "Research Progress of TOPII Inhibitors" The research prospects in the paragraph or conclusion section explain the trend of combined application of TOPII inhibitors or the possibility of combining with inhibitors of other targets. (https://doi.org/10.1002/adtp.201900202).

Response: We thank the reviewer for the valuable suggestion. The Introduction has been updated to discuss the combined application trends of TOPII inhibitors with other targeted agents, and the suggested reference has been added.

  1. When detecting apoptosis-related proteins such as PARP-1, Bax, and XIAP, only qualitative band diagrams were provided. No grayscale normalization (such as ImageJ quantitative analysis) was performed on the β-actin internal reference protein, and no statistical data from 3 independent repeated experiments were provided.

Response: In response to the reviewer’s request, we performed densitometric quantification of three independent Western blot experiments. Band intensities were analyzed using MultiGauge software (version 3.X, Fujifilm) in conjunction with the Fujifilm LAS-4000 chemiluminescence detection system. Signal intensities were normalized to the untreated control (actin), and data are presented as mean ± SD from three independent biological replicates. Statistical significance was evaluated using paired, two-tailed t-tests comparing each treatment group to its corresponding control. For transparency and reproducibility, the complete densitometric data set, including calculation tables and all bar graphs (mean ± SD with significance markers), is provided in the Supplementary Information (Excel file).

  1. Figure 4 shows that "the activity of the para-hydroxy group (8c) is higher than that of the meta-hydroxy group (8b)", but it does not explain this phenomenon in combination with molecular docking results: for example, whether the para-hydroxy group forms a stronger hydrogen bond with Gln778 at the active site of TOPIIα, or whether it improves the π-stacking interaction between the molecule and DNA bases. It only describes that "structural modification affects activity" without revealing the molecular essence of the structure-activity relationship.

Response: Thank you for your clarification. The interaction involving the OH group was amended to the manuscript and was highlighted.

  1. Although the π-interactions between 8c and DNA bases as well as the hydrogen bond with Gln778 are mentioned, the key binding sites of co-crystal ligands (such as etoposide) are not compared (e.g., the electrostatic interaction between etoposide and Asp479, and the hydrogen bond with DNA phosphate groups). Thus, it is impossible to clarify whether 8c retains the core binding characteristics of TOPII inhibitors or whether there exists a unique mode of action (such as binding to both TOPIIα and DNA simultaneously).

Response: We value the reviewer’s comment. Compound 8c occupied the same site of co-crystal ligands and involved some common interaction with TOPII amino acids and DNA bases, but due to the different functionalities, Asp479 and DNA phosphate were not involved in the interaction with 8c.

  1. In Figure 1, compound V does not list obvious target information. In compounds X and XI, "TOPO II" should be "TOP II".

Response: We thank the reviewer for the careful observation. The target information for compound V has been added, and “TOPO II” has been corrected to “TOP II” in compounds X and XI in Figure 1.

  1. In Line 151, the author mentions that the HPLC purity of all compounds is above 95%, but only the purity information of compound 8c is available in the SI.

Response: We thank the reviewer for this observation. The HPLC purity data for all compounds have now been included in the revised Supplementary Information file.

  1. The paragraph marker for Line 154 is missing.

Response: The missing paragraph marker at line 154 has been inserted in the revised manuscript. Thank you.

  1. Table 1 shows a good foundation of work, and it would be more convincing if there were SD values. The symbol appearing in the annotation (line 213) does not appear in the table; instead, there is an unannotated symbol in the table. Table 2 has the same issue.

Response: We thank the reviewer for this valuable comment. According to the NCI screening protocol, SD values are not provided as part of the standard output. However, the annotation symbols in Tables 1 and 2 have been corrected for clarity and consistency.

  1. The article lacks a discussion section.

Response: We thank the reviewer for this critique. A combined “Results and Discussion” section is already included in the manuscript, where the findings and their interpretation are presented together in accordance with the journal’s format.

  1. The structural confirmation information and HRMS of compound 8d are missing in SI. (Figure S6 and S7), 10d (Figure S22 and S23), 13C NMR of 8a and 8b
    and HRMS of 12.

Response: We appreciate the reviewer’s helpful feedback comment. The missing structural and spectroscopic data (Figures S6, S7, S22, and S23, as well as the 13C NMR spectra of compounds 8a and 8b), along with the HRMS of 12, have been added to the revised Supplementary Information.

Round 2

Reviewer 2 Report

Comments and Suggestions for Authors

I do not have other comments or concerns.